# Generalized Belief Transport

**Junqi Wang**
Department of Math & CS
Rutgers University
Newark, NJ, 07102
junqi.wang@rutgers.edu

**Pei Wang**
Department of Math & CS
Rutgers University
Newark, NJ, 07102
peiwang@rutgers.edu

**Patrick Shafto**
Department of Math & CS
Rutgers University
Newark, NJ, 07102
shafto@rutgers.edu

## Abstract

Human learners have ability to adopt appropriate learning approaches depending on constraints such as prior on the hypothesis, urgency of decision, and drift of the environment. However, existing learning models are typically considered individually rather than in relation to one and other. To build agents that have the ability to move between different modes of learning over time, it is important to understand how learning models are related as points in a broader space of possibilities. We introduce a mathematical framework, Generalized Belief Transport (GBT), that unifies and generalizes prior models, including Bayesian inference, cooperative communication and classification, as parameterizations of three learning constraints within Unbalanced Optimal Transport (UOT). We visualize the space of learning models encoded by GBT as a cube which includes classic learning models as special points. We derive critical properties of this parameterized space including proving continuity and differentiability which is the basis for model interpolation, and study limiting behavior of the parameters, which allows attaching learning models on the boundaries. Moreover, we investigate the long-run behavior of GBT, explore convergence properties of models in GBT mathematical and computationally, document the ability to learn in the presence of distribution drift, and formulate conjectures about general behavior. We conclude with open questions and implications for more unified models of learning.

Learning and inference are subject to internal and external constraints. Internal constraints include the availability of relevant prior knowledge. External constraints include the availability of time to accumulate evidence versus the need make the best decision now or environmental non-stationarity. Standard models of machine learning tend to view different constraints as different problems, which impedes development of unified learning agents.

These internal and external constraints map onto classic dichotomies in machine learning. Availability of prior knowledge maps onto the Frequentist-Bayesian dichotomy in which the latter uses prior knowledge as a constraint on posterior beliefs, while the former does not. Within Bayesian theory, a classic debate pertains to uninformative, or minimally informative, settings of priors [Jeffreys, 1946, Robert et al., 2009]. Availability of time to accumulate evidence informs the use of generative versus discriminative approaches [Ng and Jordan, 2001], and static or drift/dynamic models [Dagum et al., 1992, Murphy, 2002]. Combining constraints on probability of beliefs and costs of data models cooperative communication [Wang et al., 2020b].

Learning agents must interpolate between modes of reasoning as necessary given the constraints of the moment. Imagine observing an agent behaving in an environment. As an observer, one may wish to learn about the environment from the agent's actions. However, any inferences depend on one's model of the agent and their constraints. How is the agent updating their beliefs? Do they have stable goals, or are they changing over time? Perhaps the agent is selecting actions to communicate what they know? In order to draw inferences over these possibilities, one must parameterize the space,

37th Conference on Neural Information Processing Systems (NeurIPS 2023).

ideally in such a way one could optimize over the possibilities. Indeed, in order to implement these possibilities, the agent *themself* must parameterize the space in order to interpolate between classic dichotomies such as Bayesian and frequentist, static and dynamic environments, and helpful versus neutral agent, given constraints.

We introduce Generalized Belief Transport (GBT), based on Unbalanced Optimal Transport (Sec. 1), which paramterizes and interpolates between known reasoning modes (Sec. 2.2), with four major contributions. First, we prove continuity in the parameterization and differentiability on the interior of the parameter space (Sec. 2.1). Second, we analyze the behavior under variations in the parameter space (Sec. 2.3). Third, we study sequential learning, where learners may (not) track the empirically observed data frequencies in (Sec. 3). Fourth, we investigate predictive performance under environmental drift (Sec. 4).

**Notations.** $\mathbb{R}_{\geq 0}$ denotes the non-negative reals. Vector $\mathbf{1} = (1, \ldots, 1)$. The $i$-th component of vector $v$ is $v(i)$. $\mathcal{P}(A)$ is the set of probability distributions over $A$. For a matrix $M$, $M_{ij}$ represents its $(i, j)$-th entry, $M_{(i,\_)}$ denotes its $i$-th row, and $M_{(\_,j)}$ denotes its $j$-th column. Probability is $\mathbb{P}(\,\cdot\,)$.

# 1 Learning as a problem of unbalanced optimal transport

Consider a general learning setting: an agent, which we call a **learner**, updates their belief about the world based on observed data subject to constraints. There is a finite set $\mathcal{D} = \{d^1, \ldots, d^n\}$ of all possible data, that defines the interface between the learner and the world. The world is defined by a true hypothesis $h^*$, whose meaning is captured by a probability mapping $\mathbb{P}(d|h^*)$ onto observable data. For instance, the world can either be the environment in classic Bayesian inference [Murphy, 2012] or a **teacher** in cooperative communication [Wang et al., 2020b].

A learner is equipped with a set of hypotheses $\mathcal{H} = \{h^1, \ldots, h^m\}$ which may *NOT* contain $h^*$; an initial belief on the hypotheses set, denoted by $\theta_0 \in \mathcal{P}(\mathcal{H})$; and a non-negative cost matrix $C = (C_{ij})_{m \times n}$, where $C_{ij}$ measures the underlying cost of mapping $d^i$ into $h^j$ [1]. The cost matrix can be derived from other matrices that record the relation between $\mathcal{D}$ and $\mathcal{H}$, such as likelihood matrices in classic Bayesian inference or consistency matrices in cooperative communication (see details in Section 2.2).This setting reflects an agent's learning constraints: pre-selected hypotheses, and the relations between them and the communication interface (data set).

A learner observes data in sequence. At round $k$, the learner observes a data $d_k$ that is sampled from $\mathcal{D}$ by the world according to $\mathbb{P}(d|h^*)$. Then the learner updates their beliefs over $\mathcal{H}$ from $\theta_{k-1}$ to $\theta_k$ through a *learning scheme*, where $\theta_{k-1}, \theta_k \in \mathcal{P}(\mathcal{H})$. For instance, in Bayesian inference, the learning scheme is defined by Bayes rule; while in discriminative models, the learning scheme is prescribed by a code book.

The learner transforms the observed data into a belief on hypotheses $h \in \mathcal{H}$ with a minimal cost, subject to appropriate constraints, with the goal of learning the exact map $\mathbb{P}(d|h^*)$. We can naturally cast this learning problem as Unbalanced Optimal Transport.

## 1.1 Unbalanced Optimal Transport

Unbalanced Optimal Transport (UOT), introduced by Liero et al. [2018], is a generalization of (entropic) Optimal Transport [Villani, 2008, Cuturi, 2013, Peyré and Cuturi, 2019], that relaxes the marginal constraints. Formally, for non-negative scalar parameters $\epsilon = (\epsilon_P, \epsilon_\eta, \epsilon_\theta)$, the *UOT plan* is,

$$P^\epsilon(C, \eta, \theta) = \underset{P \in (\mathbb{R}_{\geq 0})^{n \times m}}{\arg\min} \quad \{\langle C, P \rangle - \epsilon_P H(P) + \epsilon_\eta \mathrm{KL}(P\mathbf{1}|\eta) + \epsilon_\theta \mathrm{KL}(P^T\mathbf{1}|\theta)\}. \quad (1)$$

Here, $\langle C, P \rangle = \sum_{i,j} C_{ij} P_{ij}$ is the inner product between $C$ and $P$, $H(P) = -\sum_{ij} P_{ij}(\log P_{ij} - 1)$ is the *entropy* of $P$, and $\mathrm{KL}(\mathbf{a}|\mathbf{b}) := \sum_i (a_i \log(a_i/b_i) - a_i + b_i)$ is the Kullback–Leibler divergence between vectors. It is shown in Chizat et al. [2018] that UOT plans can be solved efficiently via Algorithm 1 : Given a cost $C$, $P^\epsilon$ can be obtained by applying $(\eta, \theta, \epsilon)$-unbalanced Sinkhorn scaling on $K^\epsilon := e^{-\frac{1}{\epsilon_P} C} = (e^{-\frac{1}{\epsilon_P} C_{ij}})_{m \times n}$, with convergence rate $\tilde{\mathcal{O}}(\frac{mn}{\epsilon_P})$ [Pham et al., 2020].

---

[1]To guarantee the hypotheses are distinguishable, we assume that $C$ does not contain two columns that are only differ by an additive scalar.

**Proposition 1.** *The UOT problem with cost matrix $C$, marginals $\theta, \eta$ and parameters $\epsilon = (\epsilon_P, \epsilon_\eta, \epsilon_\theta)$ generates the same UOT plan as the UOT problem with $tC$, $\theta$, $\eta$, $t\epsilon = (t\epsilon_P, t\epsilon_\eta, t\epsilon_\theta)$ for any $t \in (0, \infty)$. Therefore, the analysis on $\epsilon$ and $t\epsilon$ are the same for general cost $C$.*

Thus a positive common factor on $C, \epsilon_P, \epsilon_\eta, \epsilon_\theta$ does not affect the solution of Eq. (1). Therefore, for the later analysis, we fix $\epsilon_P = 1$ unless otherwise stated.

**Framework: Generalized Belief Transport (GBT).** Learning, efficiently transport one's belief with constraints, is naturally a UOT problem, i.e. a *Generalized Belief Transport*. Each round, a learner, defined by a choice of $\epsilon = (\epsilon_P, \epsilon_\eta, \epsilon_\theta)$, updates their beliefs as follows. Let $\eta_{k-1}, \theta_{k-1}$ be the learner's estimations of the data distribution and the belief over hypotheses $\mathcal{H}$ after round $k - 1$, respectively. At round $k$, the learner first improves their estimation of the mapping between $\mathcal{D}$ and $\mathcal{H}$, denoted by $M_k$, through solving the UOT plan Eq. (1) with $(C, \eta_{k-1}, \theta_{k-1})$, i.e. $M_k = P^\epsilon(C, \eta_{k-1}, \theta_{k-1})$. Then with data observation $d_k$, the learner updates their beliefs over $\mathcal{H}$ using corresponding row of $M_k$, i.e. suppose $d_k = d^i$ for some $d^i \in \mathcal{D}$, the learner's belief $\theta_k$ is defined to be the row normalization of the $i$-th row of $M_k$. Finally, the learner updates their data distribution to $\eta_k$ by increment of the $i$-th element of $\eta_{k-1}$, see Algorithm 2.

---

**Algorithm 1** Unbalanced Sinkhorn Scaling

**input:** $C, \theta, \eta, \epsilon = (\epsilon_P, \epsilon_\eta, \epsilon_\theta)$, $N$ stopping condition $\omega$
**output:** $P^\epsilon(C, \eta, \theta)$
**initialize:** $\mathbf{K} = \exp(-\epsilon_P C)$, $\mathbf{v}^{(0)} = \mathbf{1}_m$
**while** $k < N$ **and not** $\omega$ **do**

$$\mathbf{u}^{(k)} \leftarrow \left(\frac{\eta}{\mathbf{K}\mathbf{v}^{(k-1)}}\right)^{\frac{\epsilon_\eta}{\epsilon_\eta + \epsilon_P}},$$

$$\mathbf{v}^{(k)} \leftarrow \left(\frac{\theta}{\mathbf{K}^T\mathbf{u}^{(k)}}\right)^{\frac{\epsilon_\theta}{\epsilon_\theta + \epsilon_P}}$$

**end while**
$P^\epsilon(C, \eta, \theta) = \mathrm{diag}(u)\mathbf{K}\mathrm{diag}(v)$

---

**Algorithm 2** Generalized Belief Transport

**input:** $C$, $\theta_0$, $\eta_0$, $h^*$, $N$, data sampler $\tau$ based on $\mathbb{P}(d|h^*)$, stopping condition $\omega$
**output:** $M, \theta$
**initialize:** $k \leftarrow 1$
**while** $k < N$ **and not** $\omega(\theta)$ **do**

$\quad M \leftarrow P^\epsilon(C, \eta_{k-1}, \theta_{k-1})$
$\quad$ get data $d^i$ sampled from $\tau$
$\quad \eta_k \leftarrow update(\eta_{k-1}, d^i)$ via update rule
$\quad \mathbf{v} \leftarrow M_{(i,\_)}$
$\quad \theta_k \leftarrow \mathbf{v}/\sum_{h \in \mathcal{H}} \mathbf{v}(h)$
$\quad k \leftarrow k + 1$
**end while**

---

## 2 Generalized Belief Transport

Many learning models—including Bayesian inference, Frequentist inference, Cooperative learning, and Discriminative learning—are unified under our GBT framework under choice of $\epsilon$. In this section, we focus on the single-round behavior of the GBT model, i.e., given a pair of marginals $(\theta, \eta)$, how different learners update beliefs with a single data observation. We first visualize the entire learner set as a cube (in terms of parameters), see Figure 1. Then, we study the topological properties of the learner set through continuous deformations of parameters $\epsilon$. In particular, we show that existing models including Bayesian inference, cooperative inference and discriminative learning are learners with parameters $(1, 0, \infty)$, $(1, \infty, \infty)$ and $(0, \infty, \infty)$ respectively in our UOT framework.

### 2.1 The parameter space of GBT model

The space of learners in GBT are parameterized by three regularizers for the underlying UOT problem (1): $\epsilon_P$, $\epsilon_\eta$ and $\epsilon_\theta$, each ranges in $[0, \infty)$. Therefore, the constraint space for GBT is $\mathbb{R}^3_{\geq 0}$, with the standard topology. When $C$, $\theta$ and $\eta$ are fixed (assume $\eta \in \mathbb{R}^m_{>0}$), the map $\epsilon = (\epsilon_P, \epsilon_\eta, \epsilon_\theta) \mapsto (P^\epsilon)$ bears continuous properties:

**Proposition 2.** [2] *The UOT plan $P$ in Equation (1), as a function of $\epsilon$, is continuous in $(0, \infty) \times [0, \infty)^2$. Furthermore, $P$ is differentiable with respect to $\epsilon$ in the interior of its domain $\mathbb{R}^3_{\geq 0}$.*

Continuity on $\epsilon$ provides the basis for interpolation between different learning agents. The proof of Proposition 2 also implies the continuity on $\eta$ and $\theta$. Further, towards the boundaries of the parameter space (where theories like Bayesian, Cooperative Communication live in), we show:

---

[2] Proofs of all claims are included in the Supplementary Materials.

**Proposition 3.** *Let $s_P$, $s_\eta$, $s_\theta \geq 0$ be arbitrary finite numbers, the following holds:*

*(1) The limit of $P^\epsilon$ exists as $\epsilon$ approaches $(\infty, s_\eta, s_\theta)$. In fact, $\lim_{\epsilon \to (\infty, s_\eta, s_\theta)} P^\epsilon_{ij} = 1$ for all $i, j$.*

*(2) As $\epsilon \to (s_P, \infty, s_\theta)$, $P^\epsilon$ converges to the solution to*

$$\min \langle C, P \rangle - s_P H(P) + s_\theta KL(P^T \mathbf{1} | \theta), \text{ with constraint } P\mathbf{1} = \eta,$$

*(3) Similarly, as $\epsilon \to (s_P, s_\eta, \infty)$, $P^\epsilon$ converges to the solution to*

$$\min \langle C, P \rangle - s_P H(P) + s_\eta KL(P\mathbf{1} | \eta), \text{ with constraint } P^T \mathbf{1} = \theta.$$

*(4) And when $\epsilon \to (s_P, \infty, \infty)$, the matrix $P^\epsilon$ converges to the EOT solution:*

$$\min \langle C, P \rangle - s_P H(P), \text{ with constraints } P^T \mathbf{1} = \theta \text{ and } P\mathbf{1} = \eta.$$

*(5) When $\epsilon \to (\infty, \infty, s_\theta), (\infty, s_\eta, \infty)$ or $(\infty, \infty, \infty)$, the limit does not exist, but the directional limits can be calculated.*

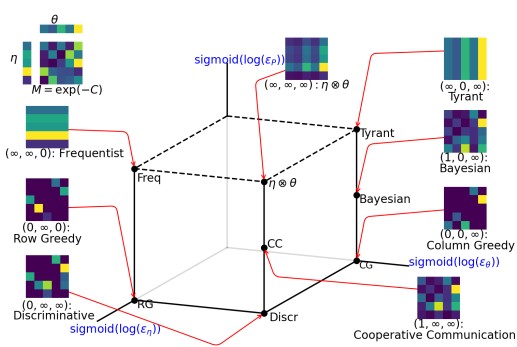

Figure 1: The parameter space $\mathcal{S}$ of GBT. Parameters $\epsilon = (\epsilon_P, \epsilon_\eta, \epsilon_\theta)$ can take the value $\infty$, rendering the corresponding regularization to a strict constraint. The two dashed edges with $\epsilon_P = \infty$ are not generally well-defined since the limits do not exist. The vertices corresponding to $\theta \otimes \eta$, Frequentist ($\eta \otimes \mathbf{1}$) and $\mathbf{1} \otimes \theta$ are the limits taken along the vertical edges. Given $(C, \theta, \eta)$ as shown in the left corner, each colored map plots each GBT learner (differ by constraints)'s estimation of the mapping between hypotheses and data (UOT plan).

The parameter space for GBT with its boundaries can be visualized in Fig. 1. Proposition 3 implies that the parameter space is $\mathcal{S} = [0, \infty]^3 \backslash (\{(\infty, \infty, x) : x \in [0, \infty]\} \cup \{(\infty, x, \infty) : x \in [0, \infty]\})$. In Fig. 1, segment $[0, \infty)$ is mapped to $[0, 1)$ by sigmoid$(\log(x))$. Then boundaries are added to the image cube $[0, 1)^3$. The dashed lines on top of the cube indicates limits that do not exist.

## 2.2 Special points in the parameter space

**Bayesian Inference.** Given observed data, a Bayesian learner (BI) [Murphy, 2012] derives posterior belief $\mathbb{P}(h|d)$ based on prior belief $\mathbb{P}(h)$ and likelihood matrix $\mathbb{P}(d|h)$, according to the Bayes rule. Intuitively, due to soft time constraint ($\epsilon_P = 1$), a Bayesian learner is a generative agent who puts a hard internal constraint on their prior belief ($\epsilon_\theta = \infty$), and omits the estimated data distribution $\eta$ in the learning process, ($\epsilon_\eta = 0$).

**Corollary 4.** *Consider a UOT problem with cost $C = -\log \mathbb{P}(d|h)$, marginals $\theta = \mathbb{P}(h)$, $\eta \in \mathcal{P}(\mathcal{D})$. The optimal UOT plan $P^{(1, \epsilon_\eta, \epsilon_\theta)}$ converges to the posterior $\mathbb{P}(h|d)$ as $\epsilon_\eta \to 0$ and $\epsilon_\theta \to \infty$. Thus, Bayesian inference is a special case of GBT with $\epsilon = (1, 0, \infty)$.*

**Frequentist Inference.** A frequentist updates their belief from data observations by increasing the corresponding frequencies of datum. Intuitively, a frequentist is an agent who puts a hard constraint on the data distribution $\eta$ ($\epsilon_\eta = \infty$), and omits prior knowledge $\theta$ ($\epsilon_\theta = 0$) in a learning process without time constraint ($\epsilon_P = \infty$). Formally we show:

**Corollary 5.** *Consider a UOT problem with $\theta \in \mathcal{P}(\mathcal{H})$, $\eta = \mathbb{P}(d)$. The optimal UOT plan $P^{(\epsilon_P, \infty, 0)}$ converges to $\eta \otimes \mathbf{1}$ as $\epsilon_P \to \infty$. Frequentist Inference is a special case of GBT with $\epsilon = (\infty, \infty, 0)$.*

**Cooperative Communication.** Two cooperative agents, a teacher and a learner, are considered in Yang et al. [2018], Wang et al. [2020b], Shafto et al. [2021]. Cooperative learners (CI) draw inferences about hypotheses based on which data would be most effective for the teacher to choose Given a data observation, a cooperative learner derives an optimal plan $L = \mathbb{P}(\mathcal{H}, \mathcal{D})$ based on a prior belief $\mathbb{P}(h)$, a shared data distribution $\mathbb{P}(d)$ and a matrix $M$ specifies the consistency between data and hypotheses (such as $M_{ij}$ records the co-occurrence of $d^i$ and $h^j$). Intuitively, a cooperative learner is also a generative agent who puts hard constraints on both data and hypotheses ($\epsilon_\eta = \infty, \epsilon_\theta = \infty$), and aims to align with the true belief asymptotically, ($\epsilon_P = 1$). Thus we show:

**Corollary 6.** *Let cost $C = -\log M$, marginals $\theta = \mathbb{P}(h)$ and $\eta = \mathbb{P}(d)$. The optimal UOT plan $P^{(1,\epsilon_\eta,\epsilon_\theta)}$ converges to the optimal plan $L$ as $\epsilon_\eta \to \infty$ and $\epsilon_\theta \to \infty$. Cooperative Inference is a special case of GBT with $\epsilon = (1, \infty, \infty)$, which is exactly entropic Optimal Transport [Cuturi, 2013].*

**Discriminative learning.** A discriminative learner decodes an uncertain, possibly noise corrupted, encoded message, which is a natural bridge to information theory [Cover, 1999, Wang et al., 2020b]. A discriminative learner builds an optimal map to hypotheses $\mathcal{H}$ conditioned on observed data $\mathcal{D}$. The map is perfect when, for all messages, encodings are uniquely and correctly decoded. Intuitively, a discriminative learner aims to quickly build a deterministic code book (implies $\epsilon_P = 0$) that matches the marginals on $\mathcal{H}$ and $\mathcal{D}$. We show that discriminative learner is GBT with $\epsilon = (0, \infty, \infty)$:

**Corollary 7.** *Consider a UOT problem with cost $C = -\log \mathbb{P}(d, h)$, $m = n$, and marginals $\theta = \eta$ are uniform. The optimal UOT plan $P^{(\epsilon_P, \epsilon_\eta, \epsilon_\theta)}$ approaches to a diagonal matrix as $\epsilon_\eta, \epsilon_\theta \to \infty$ and $\epsilon_P \to 0$. In particular, discriminative learner is a special case of GBT with $\epsilon = (0, \infty, \infty)$, which is exactly classical Optimal Transport [Villani, 2008].*

Many other interesting models are unified under GBT framework as well. GBT with $\epsilon = (0, \infty, 0)$ denotes Row Greedy learner which is widely used in Reinforcement learning community [Sutton and Barto, 2018]; $\epsilon = (\infty, \infty, \infty)$ yields $\eta \otimes \theta$ which is independent coupling used in $\chi^2$ [Fienberg et al., 1970]; $\epsilon = (\epsilon_P, \epsilon_\theta, \infty)$ is used for adaptive color transfer studied in [Rabin et al., 2014]; and $\epsilon = (0, \epsilon_\theta, \epsilon_\eta)$ is UOT without entropy regularizer developed in [Chapel et al., 2021]. Other points in the GBT parameter space are also of likely interest, past or future.

## 2.3 General properties on the transportation plans

The general GBT framework builds a connection between the above theories, and the behavior of theory varies according to the change of parameters. In particular, each factor of $\epsilon = (\epsilon_P, \epsilon_\eta, \epsilon_\theta)$ expresses different constraints of the learner. Given $(C, \theta, \eta)$ as shown in the top-left corner of Fig. 1, we plot each learner's UOT plan with darker color representing larger elements.

$\epsilon_P$ controls a learner's learning horizon. When $\epsilon_P \to 0$, a learner's UOT plan is concentrated on a clear leading diagonal which allows them to make fast decisions. This corresponding to agents who are under the time pressure of making immediate decision, i.e. discriminative learner, or row greedy learner on the bottom of the cube (Fig. 1). Most of the time, one datum is enough to identify the true hypothesis and convergence is achieved within every data observation. When $\epsilon_P \to \infty$, GBT converges to a reticent learner, such as learners on the top of the cube. Data do not constrain the true hypothesis, and learners draw their conclusions independent of the data. In between, GBT provides a generative (probabilistic) learner. When $\epsilon_P = 1$, we have Bayesian learner and Cooperative learner, for whom data accumulate to identify the true hypothesis in a manner broadly consistent with probabilistic inference, and consistency is asymptotic.

$\epsilon_\eta$ controls a learner's knowledge on the data distribution $\eta$. When $\epsilon_\eta \to \infty$, GBT converges to a learner who is aware of the data distribution and reasons about the observed data according to the probabilities/costs of possible outcomes. Examples include the Discriminative and Cooperative learners on the front of the cube. When $\epsilon_\eta \to 0$, GBT converges to a learner who updates their belief without taking $\eta$ into consideration, such as Bayesian learners on the back of the cube, and the Tyrant who does not care about data nor cost and is impossible to be changed by anybody.

$\epsilon_\theta$ controls the strength of the learner's prior knowledge. When $\epsilon_\theta \to 0$, GBT converges to learners who utilizes no prior knowledge. Hence, they do NOT maintain beliefs over $\mathcal{H}$, and draws their conclusions purely on the data distribution, such as a Frequentist learner on the left of the cube. When $\epsilon_\theta \to \infty$, GBT converges to a learner who enforces a strict prior such as Bayesian, Cooperative and Discriminative learners on the right of the cube. In particular, we show that:

**Proposition 8.** *In GBT with $\epsilon_\theta = \infty$, cost $C$ and current belief $\theta$. The learner updates $\theta$ with UOT plan in the same way as applying Bayes rule with likelihood from $P^\epsilon(C, \eta, \theta)$, and prior $\theta$.*

## 3 Sequential GBT - Static

The sequential GBT captures the asymptotic behavior of a learning problem $(C, \theta_0, h^*)$. Static world where there exists a fixed true hypotheses $h^*$ is considered in this section. Data is sampled from

$\eta = \mathbb{P}(d|h^*)$ (not necessarily related to some $h \in \mathcal{H}$). Then the learner follows GBT with cost $C$, and parameter $\epsilon$, starts with a prior $\theta_0$, then in each round $k$, applies GBT with $\eta_{k-1}$ and $\theta_{k-1}$ to generate $\theta_k$.

We investigate two cases. In the Preliminary sequential model (**PS**), we assume $\eta_k = \eta$ for all $k$. In practice, often a learner does not have access $\eta$. Instead, in each round the learner may choose to use the current observed data distribution $\eta_k(d)$ as an estimation of $\eta$, Thus we study the Real sequential model (**RS**) where $\eta_k \xrightarrow{a.s.} \eta$.

In statistics, a model is said to be consistent when, for every fixed hypothesis $h \in \mathcal{H}$, the model's belief $\theta$ over the hypotheses set $\mathcal{H}$ converges to $\delta_h$ in probability as more data are sampled from $\eta = \mathbb{P}(d|h)$. Such consistency has been well studied for Bayesian Inference since Bernstein and von Mises and Doob [Doob, 1949], and recently demonstrated for Cooperative Communication [Wang et al., 2020a]. However, the challenge arises when one tries to learn a $h^*$ that is not contained in the pre-selected hypothesis space $\mathcal{H}$. It is not clear which $h \in \mathcal{H}$ is the 'correct' target to converge to.

In this section, we demonstrate GBT's ability of learning new hypothesis. Analogize to consistency, the properties are stated directly in the language of posterior sequence $(\Theta_k)_{k=1}^{\infty}$ as random variables, focusing on whether the sequence converges (and in which sense), and how conclusive (how likely to a stable new hypothesis is learned) the sequence is.

For a learning problem $(C, \theta_0, h^*)$, results in this section are organized based on different $\epsilon_\theta$ values.

**Conclusive and Bayesian-style: $\epsilon_\theta = \infty$.** These learners are located on the right side of Cube Fig. 1. Many well-studied learners are in this class: Bayesian, Cooperative, Discriminative, Row Greedy etc. According to Prop 8, learners in this class perform "Bayesian" style learning.

When $\epsilon_\eta = 0$, i.e. the learners who update their belief without considering data distribution, (**PS**) and (**RS**) are essentially the same. The following holds:

**Theorem 9** ([Doob, 1949],[Wang et al., 2020a]). *In GBT sequential model (both (PS) and (RS)) with $\epsilon = (\epsilon_P, 0, \infty)$ where $\epsilon_P \in (0, \infty)$, the sequence $\Theta_k$ converges to some $\delta_h$ almost surely, $h$ is the closest column of $e^{-C/\epsilon_P}$ to $\eta$ in the sense of KL-divergence.*

When $\epsilon_\eta = \infty$, the models (**PS**) and (**RS**) present slightly different behaviors.

**Theorem 10** (PS). *When $\epsilon_\eta = \epsilon_\theta = \infty$, for the PS problem belief random variables of a GBT learner $(\Theta_k)_{k \in \mathbb{N}}$ converge to the random variable $Y$ in probability, where $Y = \sum_{h \in \mathcal{H}} \theta_0(h) \delta_h$ and $Y$ is supported on $\{\delta_h\}_{h \in \mathcal{H}}$ with $\mathbb{P}(Y = \delta_h) = \theta_0(h)$ for $\epsilon_\eta = \epsilon_\theta = \infty$ and $\epsilon_P \in (0, \infty)$.*

**Corollary 11.** *Given a fixed data sequence $d_i$ sampled from $\eta$, if $\theta_k$ converges to $\delta_{h^j}$, then the j-th column of $M_k$ converges to $\eta$.*

Thus a GBT learner, with access to the data distribution and using strict marginal constraints, converges to the true hypothesis mapping $\eta$ with probability 1. Moreover, the probability of which $h \in \mathcal{H}$ is shaped into $\eta$ is determined by their prior $\theta_0$. That is, GBT learners converge to the truth by reforming one of their original hypotheses into the true hypothesis.

**Proposition 12.** *When $\epsilon_\eta = \epsilon_\theta = \infty$, for the (RS) problem, the belief random variables of a GBT learner $(\Theta_k)_{k \in \mathbb{N}}$ satisfies that for any $s > 0$, $\lim_{k \to \infty} \sum_{h \in \mathcal{H}} \mathbb{P}(\Theta(h) > 1 - s) = 1$. As a consequence, $M_k$ as the transport plan has a dominant column ($h^j$) with total weights $> 1 - s$, and $|(M_k)_{ij} - \eta_k(i)| < s$.*

In fact, as long as the sequence of $\eta_k$ as random variables converges to $\eta$ in probability, the above proposition holds. The limit $\lim_{k \to \infty} \sum_{h \in \mathcal{H}} \mathbb{P}(\Theta(h) > 1 - s)$ measures how conclusive the model is.

In contrast with standard Bayesian or other inductive learners, Proposition 12 shows that a GBT learner is able to learn *any* hypothesis mapping $\eta = \mathbb{P}(d|h^*)$ up to a given threshold $s$ with probability 1. In addition to unifying disparate models of learning, GBT enables a fundamentally more powerful approach to learning by empirically monitoring the data marginal.

Fig. 2 illustrates convergence over learning problems and episodes. In each bar, we sample 100 learning problems $(C, \theta_0, h^*)$ from Dirichlet distribution with hyperparameters the vector $\mathbf{1}$. Then we sample 1000 data sequences (episodes) of maximal length $N = 10000$. The learner learns with Algo. 2 where the stopping condition $\omega$ is set to be $\max_{h \in \mathcal{H}} \theta(h) > 1 - s$ with $s = 0.001$. The

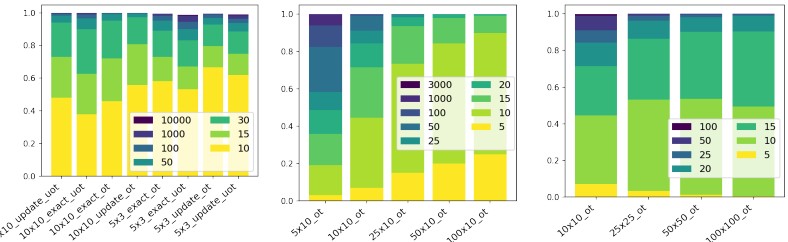

Figure 2: Evidence of general consistency: we plot the percentage of episodes that reaches a threshold (0.999) by round number (in colors of the bars). Each bar represents a size of matrix, for each bar 100 matrices were randomly sampled, and 1000 rounds were simulated per matrix. "exact" means learner uses $\eta_k = \eta$, (**PS**), "update" means learner uses statistics on current data in the episode (**RS**). "uot" takes $\epsilon = (1, 40, 40)$ and "ot" comes with exact and $\epsilon = (1, \infty, \infty)$.

$y$-axis in the plots represents the percentage of total episode converged. The color indicates in how many rounds the episode converges. For instance, in the bar corresponding to '$10 \times 10\_\text{update\_uot}$', with 10 data points (yellow portion), about $50\%$ episodes satisfy the stopping condition.

The first plot shows results for $10 \times 10$ and $5 \times 3$ matrices. The second plot shows results for rectangular matrices of dimension $m \times 10$ with $m$ ranges in $[5, 10, 25, 50, 100]$. The third plot shows results for square matrices of dimension $m \times m$ with $m$ ranges in $[10, 25, 50, 100]$. Here 'exact' and 'update' indicate the problem is (**PS**) or (**RS**), respectively. For parameters, $uot$ represents the parameter choice ($\epsilon_P = 1, \epsilon_\theta = \epsilon_\eta = 40$) vs. $ot$ represents the parameter choice ($\epsilon_P = 1, \epsilon_\theta = \epsilon_\eta = \infty$).

The first plots demonstrates that learners that do not have access to the true hypothesis (empirically builds estimation of $\eta$) learn faster than learners who have full access. The second plot indicates with a fixed number of hypotheses, learning is faster when the dimension of $\mathcal{D}$ increases. The third plot shows that the GBT learner scales well with the dimension of the problem.

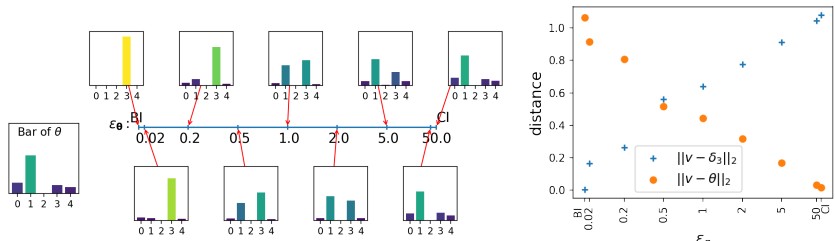

Figure 3: Left: Behavior of models spanning the line segment between BI and CI. With $\epsilon_P = 1$ and $\epsilon_\theta = \infty$, when $\epsilon_\eta$ varies from 0 to $\infty$, the theory changes from BI to CI. Each bar graphs the Monte-Carlo result of 400,000 teaching sequences, we empirically observe that the coefficients $a(h)$ of the limit in terms of $\sum_{h \in \mathcal{H}} a(h)\delta_h$ changes from BI to CI continuously from $\delta(h^3)$ by Bernstein-von Mises to $\theta_0(h)$ by Theorem 10. Right: the Euclidean distances of each coefficient $a(h)$ to BI result (blue crosses), and to CI result (orange dots).

Then we study the learners that interpolate between Bayesian and Cooperative learners (located on the line connecting CI and BI in Fig 1). Consider a fixed learning problem $(C, \theta_0, h^*)$. Consistency of Bayesian inference states that asymptotically, the learner Bayesian converges to a particular hypothesis $h_b \in \mathcal{H}$ almost surely where $h_b$ is the hypothesis closest to $h^*$ under KL divergence. Theorem 10 indicates that a GBT cooperative learner modifies one of the hypotheses into $h^*$ in probability 1. The probability of $h^j$ converges to $h^*$ is determined by $\theta_0(h^j)$.

In Fig. 3, we study the asymptotic behavior of the learners corresponding to $\epsilon = (1, \epsilon_\eta, \infty)$, with $\epsilon_\eta \in \{0, 0.02, 0.2, 0.5, 1, 2, 5, 50, \infty\}$. We sample a learning problem with a dimension $5 \times 5$ from Dirichlet distribution with hyperparameters the vector $\mathbf{1}$. Each learner $\epsilon = (1, \epsilon_\eta, \infty)$ is equipped with a fixed $C$, $\theta_0$ and $\eta_k = \eta$ for all $k$. We run $400,000$ learning episodes per learner, and plot their convergence summary in the bar graph. A continuous transition from a Bayesian learner to a cooperative learner can be empirically observed: the coefficients $a(h)$ of the limit in terms of $\sum_{h \in \mathcal{H}} a(h)\delta_h$ changes from $\delta(h^3)$ by Bernstein-von Mises to $\theta_0(h)$ by Theorem 10.

From the previous empirical results, we conclude the following conjecture:

**Conjecture 13.** *When $\epsilon = (\epsilon_P, \epsilon_\eta, \infty)$, where $\epsilon_P \in (0, \infty)$, the sequence of posteriors $\Theta_k$ from generic $C$, $\eta$, $\theta$ and $\epsilon$ as random variables satisfy* $\lim_{k \to \infty} \sum_{h \in \mathcal{H}} \mathbb{P}(|\Theta_k(h) - 1| < e) = 1$ *for any $e > 0$.*

Further, we pick out those episodes with $\theta_N(h) > 0.95$, plot the values $\mathbb{E}_{\theta_N(h) > 0.95}[\ln \theta_k(h) - \ln(1 - \theta_k(h))]$ for each $h$ against $k$ in Fig. 4. Near linear relations are observed away from the first several rounds and before the values reaches the precision threshold. These are empirical estimates of the rate of convergence.

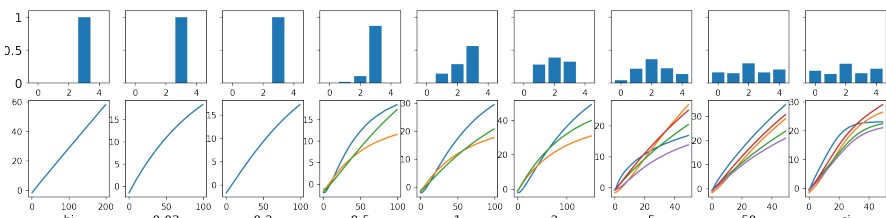

Figure 4: Top: For a learning problem $C$, behaviors of 9 different learners with $\epsilon_P = 1$, $\epsilon_\theta = \infty$ and various $\epsilon_\eta$ (denoted in figure) on conclusion distributions, $a(h)$ in bar graph, plots below bars are estimated convergence rates $\mathbb{E} \ln(\theta_k(h)/(1 - \theta_k(h)))$ averaged on episodes converging to $h$, one curve per hypothesis.

**Inconclusive and independent:** $\epsilon_\theta = 0$**.** The following holds for both (**PS**) and (**RS**):

**Proposition 14.** *For $\epsilon = (\epsilon_P, \epsilon_\eta, 0)$ with $\epsilon_P \in (0, \infty)$, as $\eta_k \to \eta$ almost surely, the sequence $\Theta_k$ of posteriors as a sequence of random variables converges in probability to variable $\Theta$, where $\mathbb{P}(\Theta = \mathbf{v}^i) = \eta(i)$ and $\mathbf{v}^i = P_{(i,\_)} / \left( \sum_{j=1}^m P_{ij} \right)$ and $P = P^\epsilon(C, \eta, \theta)$. Therefore, for any $s > 0$, $\lim_{k \to \infty} \sum_{h \in \mathcal{H}} \mathbb{P}(|\Theta_k(h) - 1| < s) = 0$ for generic (for all but in a closed subset) cost $C$ and $\eta$, $\theta$.*

With $\epsilon_\theta = 0$, the constraint on column-sum ($\epsilon_\eta$-term) fails to affect the transport plan, thus the $\Theta_k$'s in the sequence are independent from each other, in contrast that in all other cases the adjacent ones are correlated via a nondegenerate transition distribution. The independence makes the sequence of posterior-samples in one episode behave totally random, thus rarely converge as points in $\mathcal{P}(\mathcal{H})$. Furthermore, when consider the natural coupling $(\Theta_{k-1}, \Theta_k)$ from Markov transition measure for $\epsilon_\theta = 0$ (which is independent), $\mathbb{E}\left(|\Theta_{k-1} - \Theta_k|^2\right)$ converges to the variance $Var(\eta)$. In contrast, for $\epsilon_\theta = \infty$, $\mathbb{E}\left(|\Theta_{k-1} - \Theta_k|^2\right)$ converges to 0 if Conj. 13 holds.

$\epsilon_\theta \in (0, \infty)$**: partially conclusive.** From Conj. 13 and Prop. 14, together with the continuity of the transition distribution on $\epsilon$, we conjecture the following continuity on conclusiveness:

**Conjecture 15.** *For both (**PS**) and (**RS**) models, when $\epsilon = (\epsilon_P, \epsilon_\eta, \epsilon_\theta)$ with $\epsilon_P, \epsilon_\theta \in (0, \infty)$, the posterior sequence $\Theta_k$ from generated from generic $C$, $\eta$, $\theta$ and $\epsilon$ satisfy that $\lim_{k \to \infty} \sum_{h \in \mathcal{H}} \mathbb{P}(|\Theta_k(h) - 1| < s) = L$ exists, and $L \in (0, 1)$, for any $s > 0$.*

**An Exploration on Interpolation** It is popular in the state of art machine learning models that an agent learns probabilistically, but makes decisions greedily. This heuristic represents a path where a big leap on the cube was taken at the last step. An interesting question is under what circumstances this is optimal, what are the trade-offs, and under what conditions smoother trajectories are preferable. Instead of the giant leap, small steps along two paths are explored in the cube.

Our exploration takes a slightly different situation where the last step is replaced by a discriminative learning strategy $\epsilon = (0, \infty, \infty)$. We choose a matrix randomly of shape $4 \times 4$ (see Supplementary for detail), set total steps or total data points taught $N = 10$, and uniform prior $\theta \in \mathcal{P}(\mathcal{H})$ on hypotheses. Three learners are postulated: Blue performs Bayesian in first 9 steps and Discriminative in the last. Orange and Red follows two different interpolation curves with the same endpoints on the cube drawn in Fig. 5a. The curves are line and parabola segments on the cube sides.

Results of the three learners are shown in Fig. 5b. We sample $h^*$ uniformly from the 4 columns of (the column-normalized) $M$, 40000 repeats for each learner. Conclusiveness (minimal $l^1$ distance between posterior and a 1-hot vector) and posterior entropy are plotted as histograms. The results show that the smoother path may lead to a more conclusive posterior. Numerical results: Conclusiveness of Blue: mean 0.9406, standard deviation 0.1300. Conclusiveness of Orange: mean 0.9964, standard deviation

0.0327. Conclusiveness of Red: mean 0.9834, standard deviation 0.0676. Furthermore, compared with a sudden jump, gradual interpolations have lower entropy. Numerical results: entropy of Blue: mean 0.1261, standard deviation 0.2435, entropy of Orange: mean 0.0079, standard deviation 0.0629; entropy of Red: mean 0.0388, standard deviation 0.1336.

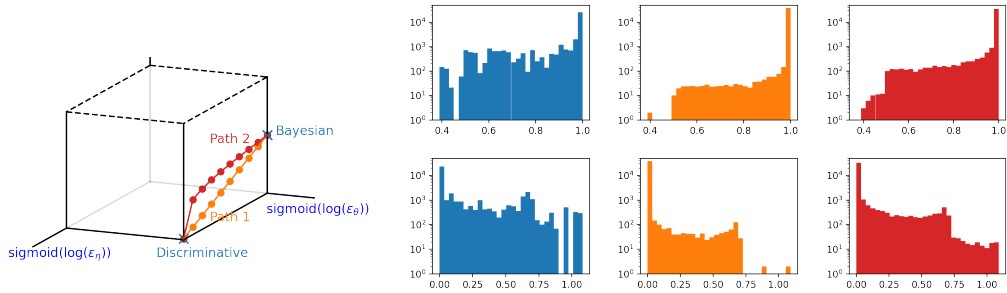

(a) Three learners          (b) Posteriors and Entropy distributions

Figure 5: (a): Baseline Blue and two learners Orange and Red following corresponding interpolation paths. (b): Results of the three learners over 40000 repeats. Top: conclusiveness, the frequency distribution of maximal posterior component. Bottom: entropy distribution.

From this experiment, in the 10-sample learning, two smoother learners behave more conclusive in their posterior with smaller posterior entropy. Meanwhile, we still know very little about the interpolation behavior, such as which path works better, how to distribute vertices on the curve, etc.

## 4 Sequential GBT - Dynamic

While static models are frequently studied, in many cases world changes dynamically. In this section, we take a first step in this direction by exploring the sequential behaviors of GBT learners assuming the world changes periodically. Demonstrate that unlike existing learners, GBT learner is capable of detecting the non-static property of a given problem.

Let integer $p > 0$ be the period, given a set of true hypotheses distributions $\vec{\eta} = \{\eta_0, \eta_1, \ldots, \eta_{p-1}\} \subseteq \mathcal{P}(\mathcal{D})$, datum $d_t$ is sampled from $d_{k\%p}$ where $k\%p$ represents the remainder of $k$ under division by $p$.

**Proposition 16.** *For a Bayesian learner, the posterior sequence $\{\Theta_i\}$ converges almost surely to the average of true hypotheses $\overline{\eta} = \frac{1}{p} \sum_{k=0}^{p-1} \eta_k$.*

For random variables of learner's posterior sequence $(\Theta_k)_{k=1}^{\infty}$, group them by period, we denote $\vec{\Theta}_t := (\Theta_{tp}, \Theta_{tp+1}, \ldots, \Theta_{(t+1)p-1})$. Here $k$ represents the time step, $t$ denotes the period index.

**Proposition 17.** *For $\epsilon$ in the interior of the cube, for (**PS**) problem, the sequence $\{\vec{\Theta}_t\}$ (random variables over $\mathcal{P}(\mathcal{H})^p$) form a time-homogeneous Markov chain. For (**RS**) problem, $\{(\vec{\Theta}_t, \frac{1}{pt} \sum_{k=0}^{p-1} t\eta_k)\}$, the random variable sequence producing samples $\{(\vec{\theta}_t, \frac{1}{pt} \sum_{k=0}^{pt-1} \delta_{d_k})\}$, forms a Markov chain.*

Next we compare different learners' behavior empirically for (**RS**) problem. For visualization, $M$ is taken of shape $3 \times 3$, thus $\mathcal{P}(\mathcal{D})$ and $\mathcal{P}(\mathcal{H})$ are both of dimension 2. If the Markov chain defined in Prop. 17 stabilizes, $\mathbb{E}[\Theta_k]$ will be periodic, matching the pattern of $\vec{\Theta}_t$. In fact, the period could be $p$, or a factor of $p$, or stabilizes where the period can be considered as 1. Thus we analyze $\vec{\eta}$ in $\mathcal{P}(\mathcal{D})$ and $\mathbb{E}[\Theta_k]$ in $\mathcal{P}(\mathcal{H})$, obtained from Monte-Carlo sampling along certain amount of episodes.

Fig. 6 (a) assume the true hypothesis travel along the triangular path connecting the 3 columns of $M$ (shown in blue crosses). We found that GBT learners with $\epsilon$ in the interior of the cube (general GBT) produce a posterior path of period $p$, while the posteriors of Bayesian and SCBI learners tend to converge (Fig. 6 (b-e)). Thus a general GBT learner can naturally detect the periodicity of the world. We simulate up to $k = 400$ steps and 10240 repeats for each learner.

Moreover, we discovered that a general GBT learner's posteriors converge to a curve whose area is proportional to the area of the path of true hypotheses. In Fig. 7, as the path of true hypotheses vary by radius and shapes, the ratio (shown as slope) between both areas tends to be the same, it is 0.1620 in (a) and 0.1616 in (d), which suggests that this ratio is independent from the path of $\vec{\eta}$.

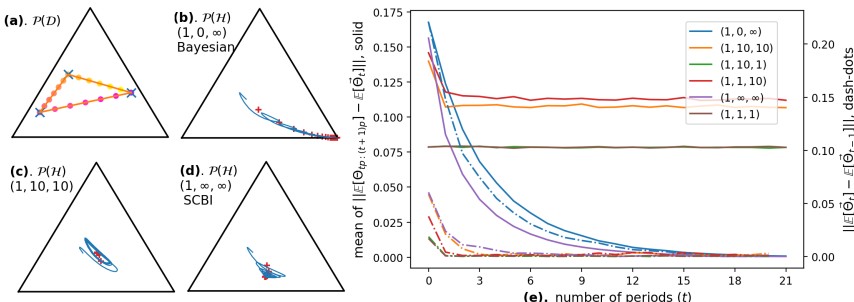

Figure 6: **(a)**. Setup: each dot represents an $\eta_k$; data are sampled along the dots from red to yellow of period 18, $M$ has 3 columns represented by the blue crosses. **(b-d)**. Bayesian, general GBT, SCBI learners, resp., blue curve shows $\mathbb{E}[\Theta_k]$ and red crosses are $\mathbb{E}[\vec{\Theta}_t]$ (mean of 18 consecutive $\mathbb{E}[\Theta_k]$'s). **(e)**. for 6 different learners (shown in colors), plot (1) the averaged distance between $\mathbb{E}[\Theta_k]$ and its center v.s. number of periods, in solid lines and left y-ticks, and (2) the step-length of $\mathbb{E}[\vec{\Theta}_t]$ between consecutive periods in dash-dots and right y-ticks.

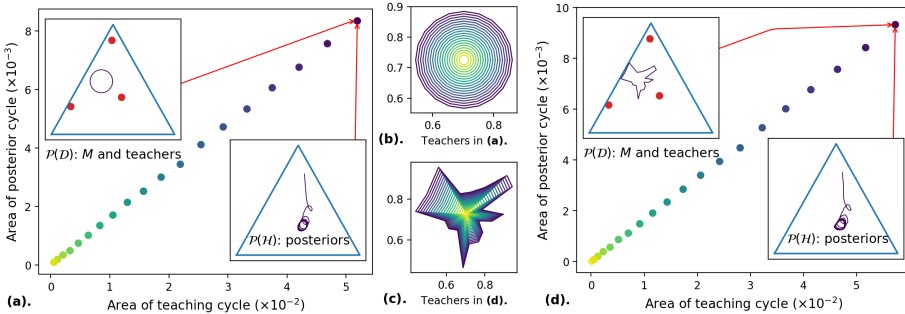

Figure 7: Behavior of a GBT learner with $\epsilon = (1, 10, 10)$ on two different paths of $\vec{\eta}$ with $p = 20$, tested in 300 steps and 10240 episodes. $M$ is fixed and represented by the red dots. Learner's posteriors form roughly periodic paths, small panels on corners of (a, d), plot path of $\vec{\eta}$ and posterior paths, ratio between their enclosed areas are shown in yellow to blue dots. (b, c) shows the 20 concentric similar paths that $\vec{\eta}$ follow. Colors are matched between paths and corresponding area ratios.

**Related Work.** Prior work defines and outlines basic properties of Unbalanced Optimal Transport [Liero et al., 2018, Chizat et al., 2018, Pham et al., 2020]. Bayesian approaches are prominent in machine learning [Murphy, 2012] and beyond [Jaynes, 2003, Gelman et al., 1995]. There is also research on cooperative learning [Wang et al., 2019, 2020b,a] see also [Liu et al., 2021, Yuan et al., 2021, Zhu, 2015, Liu et al., 2017, Shafto and Goodman, 2008, Shafto et al., 2014, Frank and Goodman, 2012, Goodman and Frank, 2016, Fisac et al., 2017, Ho et al., 2018, Laskey et al., 2017]. Discriminative learning is the reciprocal problem in which one sees data and asks which hypothesis best explains it [Ng and Jordan, 2001, Mandler, 1980]. We are unaware of any work that attempts to unify and analyze the general problem of learning in which each of these are instances.

# 5    Conclusions

We have introduced Generalized Belief Transport (GBT), which unifies and parameterizes classic instances of learning including Bayesian inference, Cooperative Inference, and Discrimination, as Unbalanced Optimal Transport (UOT). We show that each instance is a point in a continuous, differentiable on the interior, 3-dimensional space defined by the regularization parameters of UOT. Moreover, to demonstrate general GBT's capacity of supporting generalized learning, we prove and illustrate asymptotic consistency and estimate rates of convergence, including convergence to hypotheses with zero prior support, and ability of gripping dynamic of the world. In summary, GBT unifies very different modes of learning, yielding a powerful, general framework for modeling learning agents.

## Acknowledgments and Disclosure of Funding

This research was supported in part by DARPA awards HR0011-23-9-0050 and W912CG22C0001 and NSF MRI 2117429 to P.S.

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
