# OpenReview forum: "Generalized Belief Transport"
_NeurIPS.cc/2023/Conference — NeurIPS 2023 poster_

### Official Review · Reviewer_mBxm · 2023-07-05

**Soundness:** 3 good
**Presentation:** 3 good
**Contribution:** 2 fair
**Rating:** 5
**Confidence:** 1

**Summary:**

This paper describes a unifying view at various learning settings in machine learning, such as Bayesian inference, optimal communication, supervised classification and frequentist inference. All three learning settings are shown as special cases of an objective function, which can be interpreted from the lens of optimal transport.

**Strengths:**

The proposed objective function generalises such learning concepts in a general equation. This generalisation sheds new light on the various methods.

**Weaknesses:**

The current paper generalises the learning paradigms, but what new insight can be gained from this? Does this generalisation give rise to a new learning method or a more efficient version of an existing model that would solve an existing problem?

**Questions:**

* Does this work relate to Generalised Belief Propagation [1][2]? Or fit in this [3] hierarchy of inference methods?

[1] Yedidia, Jonathan S., William Freeman, and Yair Weiss. "Generalized belief propagation." NeurIPS 2000

[2] Teh, Yee, and Max Welling. "The unified propagation and scaling algorithm." NeurIPS 2001

[3] Rosen-Zvi, Michal, Michael I. Jordan, and Alan Yuille. "The dlr hierarchy of approximate inference." arXiv (2012).


**Limitations:**

-

---

> ### Author Rebuttal · Authors · 2023-08-09
>
> Thank you for your generous comments and suggestions.
> We are grateful that you find our proposed framework generalises learning concepts in a general form.
>
> * Regarding your comment on 'what new insight can be gained from this': thank you for the guiding questions on how we shall emphasize implications of GBT.
> We have organized our thoughts in the general response, please see details there.
> On the high level, our generalisation give rise to new learning model (models lie in the interior of the cube) that provide capabilities beyond existing models, which worth to be explored, and lay out a platform where interpolations between these models can be properly investigated.
>
> * Question: 'Does this work relate to Generalised Belief Propagation'?
> The provided citations are from a different perspective. Our emphasis is not on approximate inference (though there is inference, i.e. learning algorithms), but rather on formalization of the problems themselves.

---

### Official Review · Reviewer_1jqk · 2023-07-06

**Soundness:** 3 good
**Presentation:** 1 poor
**Contribution:** 2 fair
**Rating:** 4
**Confidence:** 2

**Summary:**

This work aims to unify learning approaches by specifying a 3D space where the dimensions represent modalities of learning that can be combined to specify various learning approaches, e.g., Bayesian and Frequentist approaches. The 3D space is defined by three learning constraints within Unbalanced Optimal transport (UOT).

**Strengths:**

This work's conceptual aims are quite interesting and potentially illuminating and instructive.

**Weaknesses:**

Many details are glossed over in the proofs, making them hard to read and understand. This work is limited to finite datasets and finite hypotheses, which significantly limits the generality of the work. Minimally, the hypothesis class' finiteness limits this unification's informativeness.

**Questions:**

Does this unification hold with infinite hypotheses or datasets? How might one use this unification to design new algorithms?

**Limitations:**

This work is not presented in a digestible manner; the impact of this work is not effectively communicated or demonstrated.

---

> ### Author Rebuttal · Authors · 2023-08-09
>
> Thank you for your generous comments and suggestions.
>
> * We are surprised, given the other reviewers' assessments, and disappointed that you find our paper hard to read.
> It would be helpful if you provided specifics regarding what details were glossed over. Without that information it is not possible to respond to your critique or improve the paper.
>
> * We respectfully disagree with your assessment on how finiteness limits our contributions.
>
> Rather than being a limitation, we would like to point out that our approach in fact allows generalization to continuous setting.
> GBT captures existing learning models as unbalanced optimal transport (UOT) problems.
> Theory and algorithms for continuous UOT were developed in [Chizat et al. 2018],
> which provide necessary machinery to go for continuity. We echo the reviewer that a good learner should not be limited by a pre-selected 'hypothesis class' (finite or infinite), which is a drawback of many learning model now. As detailed in general response (1a), Proposition 12 demonstrated that there are GBT learners who are able to learn hypothesis which was not in their initial hypothesis set. This potentially provides a more fundamental approach to overcome such limitation.
>
> On the other hand, for practice and implementation purposes, most if not all machine learning models are done for finite samples.
> We strongly disagree that finiteness (or not) limits the merit of this submission.
>
> * Towards question 'How might one use this unification to design new algorithms?', because the entire space of learning models is parameterized (can be viewed as a cube in Fig.1), new learning models can be explored naturally by varying the parameters.
> Existing models are mainly located on vertices of the cube, in this paper we explored two new types of learner in section 3 and section 4, please see details in the general response. For algorithm, as each model on the cube is an UOT problem, hence can be solved by any algorithm developed for UOT. The most popular approach is Sinkhorn iteration, which we included as Algorithm 1 on line 88.

---

> > ### Comment · Reviewer_1jqk · 2023-08-11
> > **Follow up to rebuttal**
> >
> > > We are surprised, given the other reviewers' assessments, and disappointed that you find our paper hard to read.
> >
> > Presumably, the authors would like this work to have an impact on the general machine learning audience who may use this unification to understand better/develop learning algorithms. I'd like to point out that reviewer NpBr noted the following:
> >
> > "The paper assumes a very high level of familiarity with concepts such as the Frequentist-Bayesian dichotomy and generative vs. discriminative approaches. Readers without extensive prior knowledge in these areas may find it challenging to fully grasp the nuances of this work. Including additional explanations or references to relevant background material would enhance the accessibility of the paper."
> >
> > I believe this is a significant limitation of the presentation.
> >
> > >We respectfully disagree with your assessment on how finiteness limits our contributions.
> >
> > I'd like to point out that the finiteness in the discussion here is not about samples, which is indeed the setting one observes in practice; it is about the finiteness of data/hypotheses that fundamentally defines the problem. The main contribution of this work is a theoretical unification of learning algorithms, so the assumption of the requirement of finiteness in both hypothesis and datasets seems very strong and unreasonable. It is stated with little discussion, even conceptually, about why it is reasonable.
> >
> > After reading your responses, other reviews, and revisiting the paper, I am happy to raise my score from 2 to 3, given that within the context of the limitations that I have pointed out, the work is sound (also increasing my score on the soundness).

---

> > > ### Author Response · Authors · 2023-08-11
> > > **Clarification**
> > >
> > > Thanks so much for following up!
> > >
> > > Clarifications of your two points:
> > > 1) Is the "high level of familiarity with concepts such as the Frequentist-Bayesian dichotomy and generative vs. discriminative" the only challenge you find with the paper? If so, that is a simple modification that would we be very happy to make. We are struggling to understand on what basis the paper deserves a "3: Reject: For instance, a paper with technical flaws, weak evaluation, inadequate reproducibility and incompletely addressed ethical considerations." given the issue is one that can be easily resolved by providing some more background. Indeed, reviewer NpBr rated the paper quite a bit higher despite raising this point.
> > >
> > > 2) To clarify, our response was about the data and hypothesis spaces. We used the phrase "finite samples" in the response which is perhaps where the confusion arose, by which we meant approximating a continuous distribution by finite samples. We don't see this as a limitation, given that most models work in finite approximations anyway. It is worth noting that there is very strong theory in OT showing convergence of the discrete to continuous when the continuous is approximated (discretized) by samples (e.g. Aude et al., 2016). This is in addition to the strong pure mathematical foundation we noted in our previous response. These results provide strong connections of the discrete case to the continuous case, and avoid some pretty heavy mathematics. We think this is a reasonable compromise, given that the discrete case is what is likely to be used, practically. We are happy to be clearer about this choice in the paper, if that is helpful.
> > >
> > > Thanks again for engaging and we are hopeful that you will consider raising your score!
> > >
> > > Aude, G., Cuturi, M., Peyre, G., and Bach, F. Stochastic optimization for large-scale optimal transport. arXiv preprint arXiv:1605.08527, 2016.

---

> > > > ### Comment · Reviewer_1jqk · 2023-08-15
> > > > **Thanks for the clarfication**
> > > >
> > > > My main two concerns remain -- the first, the accessibility of the paper, given the goal is a unification of a broad set of methods used in practice. My major concern about the finiteness, the second, remains. This works relies strongly on finite possible datasets that interface between the learner and the world; while one might only have access to a finite set of datasets in the world, they likely are not the set of all possible datasets, nor is the set that satisfies this condition likely finite. Additionally, most function classes that are used in practice are a set with infinite hypotheses -- even something as simple as a linear model.
> > > >
> > > > I think it is important to distinguish between practical constraints and the conditions necessary theory you provide for unification, which may be misaligned.
> > > >
> > > > Nevertheless, apologies for the miscalibration; after revisiting the paper further and looking at the other reviews, I agree that "a paper with technical flaws, weak evaluation, inadequate reproducibility, and incompletely addressed ethical considerations." is not an accurate score for this work", accordingly, I have raised my score "Technically solid paper where reasons to reject, e.g., limited evaluation, outweigh reasons to accept, e.g., good evaluation".

---

> > > > > ### Author Response · Authors · 2023-08-16
> > > > > **Thank you**
> > > > >
> > > > > Thanks for your continued engagement in the reviewing process. We appreciate your efforts and openness.
> > > > >
> > > > > We would love the opportunity to respond to the question of accessibility. Is there more than clearer introduction of the classic dichotomies?
> > > > >
> > > > > We stand by the relevance of finite space considerations. We could amend the paper to provide limiting analysis for countable infinities, if this would be satisfactory. The example of linear regression is a special case where analytic results are possible. In cases where analytic results are not possible, discrete (/pointwise) approximations are typical. Any continuous analysis would be sharply limited by the same considerations that prohibit analytic solutions elsewhere (computing integrals analytically is not generally possible). We opted to limit our scope to provide theory for the scope of cases with effective computational implications. We certainly understand the point about generality; we opted to narrow scope to effective generality, a choice which greatly broadens the potential audience. (No need to start from measurable spaces, etc.)
> > > > >
> > > > > Thanks again for your efforts and discussion!

---

> > > > > > ### Comment · Reviewer_1jqk · 2023-08-18
> > > > > >
> > > > > > Indeed, a clearer introduction, i.e., discussion or references, to the classic dichotomies would go a long way in accessibility. However, this is a minor point and not a hindrance in my opinion.
> > > > > >
> > > > > > I believe more discussion on the justification of finite hypothesis and defining distributions is necessary. While I understand limiting the scope of the work and not adding additional complexity to the analysis if it is not necessary, I believe it is necessary. Perhaps I am missing some nuance of the assumption, but a finite hypothesis class is very limiting, given that it is violated even for simple linear regression. For any flexible real-valued function, it is violated. In my opinion, this is a significant limitation that substantially detracts from the contribution.

---

> > > > > > > ### Author Response · Authors · 2023-08-18
> > > > > > > **Thanks again!**
> > > > > > >
> > > > > > > Thank you for your continued engagement. In the spirit of clarifying the scope of the work, and in hopes it would raise the evaluation score, would modifying the title to "Generalized Discrete Belief Transport" be helpful? We will, of course, add the suggested details in the introduction and include a discussion of the choice to work with discrete spaces emphasizing both the existence of strong approximation results that bridge the discrete and continuous and the importance of future work in the fully continuous setting.
> > > > > > >
> > > > > > > Thanks again, please let us know if there is anything else we can do to increase clarity and improve your evaluation of our work.

---

### Official Review · Reviewer_NpBr · 2023-07-06

**Soundness:** 3 good
**Presentation:** 3 good
**Contribution:** 3 good
**Rating:** 7
**Confidence:** 2

**Summary:**

Standard models of machine learning treat different internal constraints (e.g., prior knowledge) and external constraints (e.g., time availability, environmental non-stationarity) as separate problems, and thus hinder the development of unified learning agents. This paper proposes a framework called Generalized Belief Transport (GBT), which builds upon Unbalanced Optimal Transport, to unify existing learning models. GBT offers a parameterization that allows for interpolation between different modes such as Bayesian inference, Frequentist inference, cooperative learning, and discriminative learning. The authors provide theoretical analyses and empirical investigations to support their claims.

In short, the paper makes the following contributions:
- It proposes a parameterization that unifies existing learning approaches and shows its continuity and differentiability.
- It analyzes the behavior of GBT under variations in the parameter space.
- It studies sequential GPT for both static and non-static settings.


**Strengths:**

**Originality & Significance**
- The paper presents an interesting point of view that unifies various learning models with different internal and external constraints. The proposed GBT framework allows flexible combinations of $(\epsilon_P, \epsilon_\eta, \epsilon_\theta)$ and helps the development of unified learning agents.

**Quality & Clarity**
- The paper is well-structured and provides a clear explanation of the challenges and the proposed framework. The theoretical analyses conducted to establish continuity, differentiability, and behavior under parameter variations help improve the understanding of the framework. The empirical investigations on sequential learning and predictive performance under environmental drift also demonstrate the practical applicability of GBT. The authors provide the proofs for the key theorems and the code for reproducing experiments in the supplementary material.


**Weaknesses:**

**Empirical significance**
- It is very interesting to unify seemingly different learning models like the Frequentist and the Bayesian. However, the paper touches vaguely on the practical implications of such a unified framework. For instance, are there trade-offs for taking different points in the parameter space? How does the proposed framework extrapolate? Does it lead to new learning methods that are previously under-explored? What are some practical future steps for improving/utilizing the proposed framework?


**Presentation**
- The paper assumes a very high level of familiarity with concepts such as the Frequentist-Bayesian dichotomy and generative vs. discriminative approaches. Readers without extensive prior knowledge in these areas may find it challenging to fully grasp the nuances of this work. Including additional explanations or references to relevant background material would enhance the accessibility of the paper.

**Other details**
- Section 1: missing definition of $\eta$ (estimation of the data distribution) in the first 4 paragraphs.
- The arrangement of the color bars in Figure 2 (left) can be more organized. Currently it’s hard to read and distinguish the setting names.
- Color legend is missing for Figure 3.


**Questions:**

- How is the area for the GBT learner’s posteriors measured for Figure 6 (the curves do not seem to be closed)?

- Are there any existing efforts to combine discriminative learning methods with Bayesian approaches (class priors) to tackle a learning problem? How does the proposed framework handle such integration?


**Limitations:**

Given the lack of previous work attempting to unify and analyze the general problem of learning under constraints, what are the potential research directions and open questions in this area? How can the field benefit from a unified framework for learning and inference?

---

> ### Author Rebuttal · Authors · 2023-08-09
>
> Thank you for your constructive suggestions and insightful comments.
> We are excited that you find our proposed 'framework allows flexible combinations of and helps the development of unified
> learning agent'. Please see our answers to your questions below.
>
> * Empirical significance and limitations: thank you so much for the insightful comment on how we shall improve our presentation on these points. We have organized our thoughts in the general response, please see details there.
>
> * Presentation:  as suggested, we will add a preliminary section on existing learning models prior introduce GBT framework (around line 63, prior section 1.1). Thank you for the detail comments, they will also be addressed in revision.
>
> * Question: 'How is the area for the GBT learner’s posteriors measured for Figure 6 (the curves do not seem to be closed)?'
>
> * Answer: You are right, the entire curve is not closed. To calculate the area, we plotted the mean posterior data from round 1 to round 300 on $\mathcal{P}(\mathcal{D})$ (the unilateral triangle),
> and observed that these points become periodic of size 20 (the period of the true hypothesis) as data increase. Thus we divided the mean posterior data from round 1 to round 300 into 15 full periods of length 20.  For each period, consider the polygon spanned by the 20 points, its shape stabilized after initial periods. Hence, we used average areas of last a few periods as the area of GBT learner’s posteriors.

---

> > ### Comment · Reviewer_NpBr · 2023-08-11
> >
> > Thank you for the response. My concerns are addressed. I'm happy to keep my rating.

---

> > > ### Author Response · Authors · 2023-08-11
> > > **Follow up**
> > >
> > > Thank you!

---

### Official Review · Reviewer_7Pt6 · 2023-07-06

**Soundness:** 3 good
**Presentation:** 3 good
**Contribution:** 3 good
**Rating:** 6
**Confidence:** 3

**Summary:**

The authors introduce the concept of Generalized Belief Transport to unify and parameterize 3 different axes of learning from within the formalism of Unbalanced Optimal Transport.  Corresponding limits points on the "cube" of these 3 axes recapitulate many common learning paradigms from the literature (e.g., bayesian inference, frequentist inference, cooperative communication, etc.).  The authors further demonstrate an algorithm for solving some synthetic instances of these various problems, and demonstrate various tradeoffs as one moves from one type of learning to another.

**Strengths:**

Originality:  The connections between these formalisms through the lens of UoT (and the proofs) are certainly novel.

Quality: The overall quality of presentation is high, and I particularly like the "cube" device for visualizing the different limit points.

Clarity: The presentation is incredibly straightforward, and the proofs are quite clear.

Significance: This is less clear, but a more general formalism that can automatically facilitate many different, familiar forms of learning seems instrumentally valuable for further methods development.


**Weaknesses:**

How useful this formalism will _actually_ be for practitioners is my primary point of uncertainty.  Typically, any given practitioner doesn't really have any doubt about where they are on the "cube" for their learning problem, and thus they throw the most powerful technique available to them at that particular point of learning-agent-space.  This work might be better served by a less synthetic example, i.e., an example problem that _requires_ moving between points in learning-agent-space, to which this formalism would actually be uniquely well suited.  But a good (better yet, a *convincing*) example of such a thing is not obvious to me.  tldr: how do I actually use this formalism, if I know exactly where I am in learning agent space, and why wouldn't I use one of the more well-known tools?

**Questions:**

see above

---

> ### Author Rebuttal · Authors · 2023-08-09
>
> Thank you for your generous comments and suggestions.
> We are grateful that you find our utilization of UOT is novel and our presentation is clear and straightforward.
>
> Most importantly, your request for a *convincing* example hit home with us. The proposed addition to the introduction is, we believe, convincing. We would very much appreciate your thoughts on that example, and hope you agree!
>
> Regarding your concerns on `how useful this formalism will actually be for practitioners',
> please see detailed clarification in the general response.
> On the high level, we believe (1) models lie in the interior of the cube provide capabilities beyond existing models, which worth systematic exploration.
> (2) more importantly, rather than create a new model for the modeler, GBT is developed from agents point of view (thank you for pointing this out, we will add a clarification in revision).
> In practice, agents do not know the situation well enough to identify an optimal model in the cube at the beginning of a mission.
> They need to find their way towards the optimal model as data come in. The cube represents a space of possible ways agent may learn about the world.  Such interpolation can be done by gradually changing the agent's $\epsilon$ as shown on Page11 Fig 1 of supplementary material. Algorithms that effectively (and ideally efficiently) do this inference based on observed data is an important direction for future work.

---

> > ### Comment · Reviewer_7Pt6 · 2023-08-14
> >
> > I thank the author's for their response, and have raised by score to a 6 in response.

---

> > > ### Author Response · Authors · 2023-08-15
> > > **Thanks!**
> > >
> > > Thank you! Please let us know if there are any further questions or clarifications we can provide! We would be happy to provide further details.

---

### Official Review · Reviewer_vUNg · 2023-07-25

**Soundness:** 3 good
**Presentation:** 2 fair
**Contribution:** 2 fair
**Rating:** 5
**Confidence:** 1

**Summary:**

This paper proposes a framework called Generalized Belief Transport that unifies different types of machine learning models. The authors have proven a number of properties about their proposed framework.

**Strengths:**

* The problem proposed seems relevant given the different number of approaches to machine learning. If successful the proposed framework can greatly unify prior work.

**Weaknesses:**

* I do not have any prior knowledge about the problem studied in this paper at all, so I found the paper to be difficult to understand. That might not necessarily be the fault of the authors, but I will defer to the experts in this area to comment.

**Questions:**

N/A

---

> ### Author Rebuttal · Authors · 2023-08-09
>
> Thank you for your comments, please see further clarification on the practical implications in the general response. We are hopeful that you will appreciate the importance of this work!

---

### Author Rebuttal · Authors · 2023-08-09

We would like to thank all reviewers for their generous comments and suggestions.
Here we clarify the practical implications of proposed framework.
Specific comments are addressed for each reviewer separately.

We are encouraged that all reviewers agree that Generalized Belief Transport (GBT) establishes a uniform mathematical foundation for a broad class of learning models, upon which, basic questions in learning can be answered rigorously.

* We agree with Reviewer 7Pt6 "This work might be better
served by a less synthetic example, i.e., an example problem that requires moving
between points in learning-agent-space, to which this formalism would actually be uniquely well suited".
Here is a concrete example:

``Suppose a learner observing an *agent* behaving in an environment. As an observer, one may wish to learn about the environment from the agent's actions. However, any inferences one draws depend on beliefs about the agent. How is the agent updating their beliefs? Do they have stable goals, or are they changing over time? Perhaps the agent is selecting actions to communicate what they know? In order to draw inferences over these possibilities, one must parameterize the space, ideally in such a way one could optimize over the possibilities. Indeed, with such a framework, one would be able to naturally interpolate between classic dichotomies such as Bayesian and frequentist, static and dynamic environments, and helpful versus neutral agents. We propose such a framework. ''

We will insert this text in the introduction between the second and third paragraphs, to help the reader understand the need for our generalized framework.

* Towards answering "what are the potential research directions and open questions in this area?
How can the field benefit from a unified framework for learning and inference? what
new insight can be gained from this?" (Reviewer NpBr and Reviewer mBxm),
here are our thoughts:


(1) Because the entire space of learning models is now parameterized (can be viewed as a cube in Fig.1),
new learning models can be explored naturally,
or even optimized with respect to particular tasks in the feature.
Existing models are mainly located on vertices of the cube, in this paper we take a first probe into the cube with two cases below.
Full exploration of the cube and an effective optimization routine are left to future work, to avoid over complicating an already technically detailed paper.

(a) In section 3,  we proved that there are learners in the cube can learn a new hypothesis naturally.
A drawback of Bayesian inference is that only hypotheses in the original hypothesis set can be learned.
Proposition 12 showed that GBT learners with $\epsilon_{\eta} = \epsilon_{\theta} = \infty, \epsilon_P\in (0,\infty)$ are able to learn *any* hypothesis, which begins to approximate the flexibility of human learning.

(b) In section 4, we demonstrated learners in the cube who are capable of learning in dynamic environments (Fig.5 and Fig.6).
To be more concrete (as suggested by Reviewer 7Pt6), consider the situation where a learner observes data from a ground truth that is dynamic.
For example the weather gradually changes, the climate slowly drifts over time, learners learn with experience, etc.
Figure 5 shows when the ground truth travel along a triangular path (Fig.5a),
Bayesian learner converges to a fixed hypothesis on a vertex (Fig.5b) whereas GBT learner with parameters $(1, 10, 10)$ was able to detect there is a cyclic pattern (Fig.5c).


(2) The differentiability of GBT paves the path for online interpolation between learning models.
Here are several cases where movement in the space yields interesting, novel theory:

(a) It is popular in the state of art machine learning models that an agent learns probabilistically,
but makes decisions greedily.
This heuristic represents a path where a big leap on the cube was taken at the last step.
An interesting question is under what circumstances this is optimal, what are the trade-offs, and under what conditions smoother trajectories are preferable.

(b) When we communicate with others, human learners can move from Bayesian inference to cooperative communication gradually,
which involves recognizing that the people select data purposefully, rather than sampling at random, conditional on the hypothesis they wish to convey.
Such smooth interpolations can be achieved by modifying a learner's location on the cube as demonstrated in Section 3.1 (page 11 Fig.1) in the supplement materiel.
As Reviewer 7Pt6 pointed out most of existing research focus on the case where "practitioner doesn't really have any doubt about where they are on the cube".
However, in reality, both environment and other agents are dynamically changing,
we believe GBT can facilitate research in the direction of building a learner who is able to adopt appropriate learning models based on incoming data,
rather than learning in a fixed model.

---

### Decision · Program_Chairs · 2023-09-21

**Decision:**

Accept (poster)

**Comment:**

This paper proposes a framework called Generalized Belief Transport that unifies different types of machine learning models and allows for interpolation between different modes such as Bayesian inference, Frequentist inference, cooperative learning, and discriminative learning.

The authors further demonstrate an algorithm for solving some synthetic instances of these various problems and demonstrate various tradeoffs as one moves from one type of learning to another.

Some reviewers found to paper difficult to understand, while others commented that the “presentation is incredibly straightforward, and the proofs are quite clear” and the paper “provides a clear explanation of the challenges and the proposed framework”

A weakness of the paper is that it touches only vaguely on the practical implications of the proposed model.